# Interrelations between Patients’ Clinicopathological Characteristics and Their Association with Response to Immunotherapy in a Real-World Cohort of NSCLC Patients

**DOI:** 10.3390/cancers13133249

**Published:** 2021-06-29

**Authors:** Ana Callejo, Joan Frigola, Patricia Iranzo, Caterina Carbonell, Nely Diaz, David Marmolejo, Juan David Assaf, Susana Cedrés, Alex Martinez-Marti, Alejandro Navarro, Nuria Pardo, Ramon Amat, Enriqueta Felip

**Affiliations:** 1Clinical Research Department, Vall d’Hebron Institute of Oncology (VHIO), Passeig Vall d’Hebron 119-129, 08035 Barcelona, Spain; acallejo@vhio.net (A.C.); piranzo@vhio.net (P.I.); ndiaz@vhio.net (N.D.); jassaf@vhio.net (J.D.A.); scedres@vhio.net (S.C.); amartinezmarti@vhio.net (A.M.-M.); anavarro@vhio.net (A.N.); npardo@vhio.net (N.P.); 2Oncology Department, Vall d’Hebron University Hospital, Passeig Vall d’Hebron 119-129, 08035 Barcelona, Spain; dmarmolejo@vhebron.net; 3Thoracic Cancers Translational Genomics Unit, Vall d’Hebron Institute of Oncology (VHIO), C/Nazaret 115-117, 08035 Barcelona, Spain; jfrigola@vhio.net (J.F.); ccarbonell@vhio.net (C.C.)

**Keywords:** immune checkpoint inhibitors, NSCLC, immunotherapy, immune related adverse events, LDH, biomarkers

## Abstract

**Simple Summary:**

Immunotherapy and, in particular, immune checkpoint inhibitors (ICIs), have transformed non-small cell lung cancer treatment options. Although many patients are treated with ICIs, a large number do not respond. Thus, there is a need to identify biomarkers of response. In this study, we evaluated the value of multiple routinely collected variables (from metastatic sites to levels of several conventional peripheral blood parameters) as biomarkers of response to ICIs. Our data indicates that, although several characteristics are associated with response, many present strong interrelations, which should be taken into account when creating compendiums of biomarkers to maximize their predictivity. Finally, we describe a collection of characteristics (LDH levels, sex, and presence or absence of immune related adverse events) that, in our group of patients, had the best predictive value.

**Abstract:**

Immune checkpoint inhibitors (ICIs) have transformed non-small cell lung cancer (NSCLC) treatment. Unfortunately, only some patients benefit from these therapies. Thus, certain clinicopathological characteristics of the patients have been proposed as biomarkers of ICIs response. We assembled a retrospective cohort of 262 NSCLC patients treated with ICIs, compiled relevant clinicopathological characteristics, and studied their associations with treatment outcome using Cox proportional-hazards survival models. Additionally, we investigated the interrelations between clinicopathological features and devised a method to create a compendium associated with ICIs response by selecting those that provide non-redundant information. In multivariate analyses, ECOG performance status (hazard ratio (HR) 1.37 (95% CI 1.11 to 1.68), *p* < 0.005), LDH (HR 1.24 (95% CI 1.03 to 1.48), *p* = 0.02)) and PD-L1 negativity were associated with decreased progression-free survival (PFS) (HR 1.92 (95% CI 1.03 to 3.58), *p* = 0.04), whereas presentation of immune-related adverse events (irAEs) (HR 0.35 (95% CI 0.22 to 0.55, *p* < 0.005) or females (HR 0.52 (95% CI 0.33 to 0.80, *p* < 0.005) had longer PFS. Additionally, numerous clinicopathological indicators were found to be interrelated. Thus, we searched for features that provide non-redundant information, and found the combination of LDH levels, irAEs, and gender to have a better association with ICIs treatment response (cross-validated c-index = 0.66). We concluded that several clinicopathological features showed prognostic value in our real-world cohort. However, some are interrelated, and compendiums of features should therefore consider these interactions. Joint assessment of LDH, irAEs, and gender may be a good prognostic compendium.

## 1. Introduction

Immune checkpoint inhibitors (ICIs) have become fundamental to treat advanced non-small cell lung cancer (NSCLC) patients in recent years and are now included as a standard of care. Nevertheless, many patients do not exhibit benefit derived from these treatments. Hence, the biomedical community has intensively attempted to identify biomarkers of response. A tumor’s molecular features, such as Programmed Death-Ligand 1 (PD-L1) expression or tumor mutational burden (TMB), are associated with positive response to ICIs and remain the most prominent biomarkers [1,2,3,4]. In addition, different clinical parameters have been studied as prognostic or predictive markers of response to ICIs (individually or as a compendium). As an example, peripheral blood markers, such as the neutrophil-to-lymphocyte ratio (NLR), and other potential inflammatory biomarkers, have been associated with ICIs response [5,6,7,8,9].

However, despite the effort to identify predictive biomarkers, the use of these parameters has not yet been established to discriminate patients who may not receive ICIs because of the lack of benefit.

Notably, treatment with ICIs may cause a wide range of immune-related adverse events (irAEs) that can compromise treatment continuation and, in some cases, threaten a patient’s life. Interestingly, few studies have associated irAEs with better response and overall survival in NSCLC. This is likely due to the fact that these undesired side-effects are intrinsically related to the ICIs’ mechanism of action, and thus appearance of irAEs may be a surrogate of its effectiveness [10,11,12,13].

Because ICIs treatment is widely used, cohorts with a larger number of patients evaluating multiple clinicopathological variables in real-world settings can be assembled. The study of these variables can become a useful tool to investigate unselected large patient datasets, and associations between ICIs response (and irAEs), baseline clinical variables and concomitant medications, and to assess their utility as prognostic and predictive biomarkers. Outcomes from these real-world studies can support the medical oncologist community and potentially aid clinicians in their daily decision-making process. 

In this study, we retrospectively collected clinicopathological information from a cohort of ICIs-treated NSCLC patients, studied the interplay between these clinicopathological features, and associated them with clinical outcomes. Additionally, we designed a method to generate a compendium of features that maximizes the prognostic value while minimizing the number of parameters included.

## 2. Materials and Methods

### 2.1. Cohort Characteristics

We conducted a real-world, unicentric, retrospective, observational study to determine the interplay between clinical outcomes in advanced NSCLC patients treated with ICIs and different features (clinicopathological, demographic, and treatment-related).

Inclusion criteria were: age ≥ 18 years; diagnosis of metastatic NSCLC; treatment with anti PD-1 or anti PD-L1 at any treatment line (at least one cycle administered of ICIs). Exclusion criteria were other concomitant malignancies or loss of follow-up.

We investigated 262 consecutive patients diagnosed with advanced NSCLC and treated with ICIs as a first or subsequent line at Vall d’Hebron University Hospital between January 2013 and December 2019. The data source was digital clinical records of the patients, selected by type of treatment, and the data collection was performed using Microsoft Excel, anonymizing personal information. Patients’ characteristics, such as age, sex, performance status, comorbidities, concomitant treatment, and smoking history, were recorded. Similarly, cancer-related features, such as histology, molecular status, PD-L1 expression (in tumor cells), stage at diagnosis, and metastases localization were monitored (bone, liver, and central nervous system were recorded prior to ICIs treatment). Additionally, we recorded other variables: blood biomarkers prior to ICIs treatment (platelets-to-lymphocyte ratio; neutrophil-to-lymphocyte ratio and LDH levels); number of previous systemic treatments; and immune-related adverse events (defined according to Common terminology criteria for adverse events (CTCAE) version 5.0. Department of Health and Human Services, National Institutes of Health, National Cancer Institute). (https://ctep.cancer.gov/protocoldevelopment/electronic_applications/docs/ctcae_v5_quick_reference_5x7.pdf, accessed on November 2017).

We used progression-free survival (PFS) and overall survival (OS) to measure clinical outcomes. Patients’ outcomes were evaluated by radiological assessment in clinical practice, i.e., with a CT scan each 8–14 weeks according to RECIST criteria V1.1 [14]. The data cut-off period was December 2019.

### 2.2. Statistical Analysis

#### 2.2.1. Cohort Pre-Processing

For 14 of the 262 patients the reason for the termination of ICIs treatment was not available. Therefore, these patients were not included in the survival analyses.

#### 2.2.2. Age Stratification

Patients were stratified into three age categories: those under 55 years old were labeled as “young”, those between 55–75 years old were considered “middle”, and those over 75 years old were classified as “elder”. 

#### 2.2.3. PD-L1 Pre-Processing

In those patients in which the PD-L1 assessment was qualitative, it was transformed into “positive” or “negative”. Thus, patients with a PD-L1 assessment of 0 were considered to be “negative”, whereas patients with any value greater than 0 were considered to be “positive”. 

#### 2.2.4. PFS Computation

PFS was computed as the number of days between the start of the first line of ICIs administered to the patient and the end of treatment. When treatment was terminated due to progression, the event was considered to have occurred. If treatment termination was due to other reasons (e.g., toxicity or protocol), the patient was censored at the time of ICIs termination. 

#### 2.2.5. OS Computation

For deceased patients, the OS was computed as the number of days between the beginning of the first line of ICIs and the exitus date, and the event was considered to have occurred. For the remainder of the patients, the OS was computed as the number of days from the start of the first line of ICIs and the date of the patient’s last follow-up. The OS of these patients was censored at this date.

#### 2.2.6. Survival Models

Univariate and multivariate Cox proportional-hazards models were built using the lifelines Python library (doi:10.5281/zenodo.4579431) with a step-size parameter set to 0.5. Categorical variables were transformed into dummy variables, and numerical variables were standardized.

#### 2.2.7. Interactions between Features

All features were combined in a pairwise manner. When both features were numeric, their association was assessed via Pearson regression. When both features were categorical, their association was assessed using a chi-squared test. When one feature was categorical and the other numerical, a Kruskal–Wallis test was performed.

#### 2.2.8. Feature Selection

A Cox proportional-hazards survival model for each feature was performed using patients’ PFS as a measure of ICIs benefit. To evaluate the predictive potential of each model, 3-fold cross validation was performed using the k_fold_cross_validation function of the lifelines Python library. This process was repeated 10 times for each feature, using seeds from 0 to 9 defined using the seed function of the random Python library. Thus, 3-fold cross-validation performed 10 times led to 30 Cox proportional-hazards survival models per feature. For each of these models, the Harrell’s Concordance Index was retrieved and the mean concordance per feature was computed, and the feature with the highest mean concordance was selected. Then, additional models with this feature combined with the remainder in a pairwise manner were built, again performing a 3-fold cross-validation 10 times and retrieving the concordance index. Thus, the feature that combined with the previously selected feature led to the greatest increase in concordance, and was selected and included in the model. This procedure was repeated iteratively until no feature led to a concordance increase of at least 0.01.

## 3. Results

### 3.1. Description of the Cohort

Our cohort included a total of 262 advanced NSCLC cancer patients. Their characteristics are summarized in Table 1. The median age at the moment of ICIs treatment initiation was 63.7 years old (range: 34.0–87.0), with a proportion of males of 69%. Smokers constituted 88% of our cohort. The patients had a median PFS of 5.6 months and a median OS from the start of the ICIs treatment of 11.1 months. A total of 203 patients were treated with ICIs as monotherapy (median PFS 5.6 months), 40 with ICIs combined with chemotherapy (median PFS 8.5 months), and 19 with ICIs combined with tyrosine kinase inhibitors (median PFS 4.7 months). No statistical differences were observed between the three groups (ICIs vs. TKI + ICIs, *p* = 0.69; ICIs vs. CT + ICIs, *p* = 0.26; TKI + ICIs vs. CT + ICIs, *p* = 0.58)).

### 3.2. Association of Clinical Features with Clinical Outcome

Univariate and multivariate Cox proportional-hazards models revealed that multiple clinicopathological features were associated with ICIs clinical outcome (PFS and OS) (Figure 1, Table 2 and Table 3, Appendix A). 

In a multivariate analysis, ECOG performance status (HR 1.37 (95% CI 1.11 to 1.68), *p* < 0.005), higher levels of LDH (HR 1.24 (95% CI 1.03 to 1.48), *p* = 0.02)) and PD-L1 negativity (HR 1.92 (95% CI 1.03 to 3.58), *p* = 0.04)) were associated with decreased PFS. The presence of metastasis in different sites (prior to ICIs treatment) was also included, with bone metastasis being the only one associated with shortened PFS (HR 1.47 (95% CI 1.0 to 2.17), *p* = 0.05). By comparison, smoking habit, considered as a continuous variable (packs-year), was associated with increased PFS (HR 0.78 (95% CI 0.62 to 0.98), *p* = 0.03)), but not when considered as a categorical feature (smoker/non-smoker) (smoker, HR 1.38; 95% CI 0.7 to 2.75, *p* = 0.36). Female gender was also associated with extended PFS (HR 0.52 (95% CI 0.33 to 0.80, *p* < 0.005). Finally, patients exhibiting irAEs had substantially longer PFS (HR 0.35 (95% CI 0.22 to 0.55, *p* < 0.005). None of the other parameters were statistically significantly associated with ICIs PFS. 

Most of the features associated with decreased PFS were also associated with diminished OS in a multivariate analysis: ECOG performance status (HR 1.39 (95% CI 1.12 to 1.74), *p* < 0.005), LDH levels (HR 1.23 (95% CI 1.01 to 1.48), *p* = 0.04), and bone metastasis (HR 1.94 (95% CI 1.29 to 2.92), *p* < 0.005), whereas PD-L1 negativity was nearly statistically significant (HR 2.08 (95% 0.99 to 4.35), *p* = 0.05). Additionally, neutrophil-to-lymphocyte (NLR) ratio, as a continuous variable (HR 1.46 (95% CI 1.02 to 2.09), *p* = 0.04) and the presence of liver metastasis (HR 1.84 (95% CI 1.13to 3.00), *p* = 0.01) were also associated with decreased OS. Notably, from the features associated with increased PFS, only irAEs was also associated with longer OS (HR 0.31 (95% CI 0.18 to 0.52), *p* < 0.005). 

Concomitant medications were also included in our analysis. Patients treated with statins exhibited extended OS (HR 0.59 (95% CI 0.37 to 0.96), *p* = 0.04). By comparison, patients who received corticoids prior to ICIs treatment exhibited lower OS, although this only reached statistical significance in the univariate analysis (HR 1.58 (95% CI 1.03 to 2.43, *p* = 0.04). 

None of the other variables analyzed (i.e., histology, line of treatment, age group) were associated with PFS or OS in the multivariate analysis (Appendix A). Nevertheless, large cell neuroendocrine tumors were associated with decreased PFS in a univariate model (Appendix A). Similarly, patients with a higher platelets-to-lymphocyte ratio (PLR) exhibited curtailed PFS (Appendix A). 

### 3.3. Interactions between Features and Proposed Compendium

The differences observed between the univariate and multivariate analyses could be due to the fact that some of the features included in the analyses have a high degree of collinearity. Thus, we decided to study their putative associations (Appendix A). Unsurprisingly, many features were found to be associated with each other. For instance, ECOG performance status was associated with LDH levels, NLR, or central nervous system (CNS) metastasis, and the latter was associated with liver metastasis or corticoids treatment. A full description of interactions can be found in Appendix A. 

Because many features interact with each other, the combination of some of these features in a compendium may not improve their prognostic value if they provide redundant information. Therefore, we devised a method to create a compendium of features that would maximize the predictive value, while using the minimum number of features. For this purpose, we begun by selecting the clinical feature that, when incorporated in a Cox proportional-hazards model (with a 3-fold cross validation iteratively performed 10 times), led to a higher concordance index. Next, we combined this feature with the remaining features in a pairwise manner, constructed new Cox proportional-hazards models, and selected the feature that led to a greater increase in the concordance index in combination with the previously selected feature, and repeated this step iteratively by adding new features to the model until none of them increased the concordance index by more than 0.01. Thus, we observed that the combination of LDH levels, irAEs, and gender offered the best prognostic value based on PFS (Figure 2A). The impact of varying these features, given our models, can be visualized using a partial effects representation (Figure 2B). It should be highlighted that similar results were obtained when considering only patients treated with ICIs as monotherapy (77% of our cohort) (cross-validated c-index = 0.67). Importantly, these three features were also predictors of OS (cross-validated c-index = 0.665).

## 4. Discussion

In this study, we assembled a large cohort of advanced NSCLC patients treated with ICIs at the Vall d’Hebron Hospital, and integrated relevant clinicopathological data to investigate their prognostic value, using both PFS and OS as measurements of treatment benefit. Additionally, we studied the interplay between features, which showed that some of the features that were associated with response to ICIs strongly correlate with each other. Taking this into account, we devised a method to generate a compendium of features (LDH, irAEs to ICIs, and gender) that, in combination, better associate with PFS, and retain their prognostic value when considering OS.

We analyzed different routinely assessed molecular and cellular peripheral blood markers, and studied them in the context of ICIs. NLR was first described as prognostic marker of OS [5], which we also observed in our cohort. Nevertheless, no association between NLR and ICIs response was observed when considering PFS. Higher LDH levels, when considered as a continuous variable, were associated with worse PFS and OS. This was similar to the findings of other studies, which used cut-off points to transform LDH levels into a categorical variable [6,7,15]. Among all other blood-based parameters considered in our study, PLR was also associated with OS, but only in the univariate model, perhaps indicating that its association with OS is driven by its correlation with other features, for instance, NLR (Appendix A) [16,17]. 

We also incorporated information regarding different concomitant medications into our survival models. Notably, statins statistically significantly associated with OS but not with PFS, indicating that treatment with statins is likely a prognostic marker rather than a predictor of ICIs response. Recent data from a pan-cancer cohort found that statins are not associated with PFS nor OS [13,18]. Nevertheless, in their OS multivariate analysis, statins treatment was nearly statistically significant (*p* = 0.06); thus, it is possible that statins might have a different prognostic value across different tumor types. Corticoids are one of the most well-studied concomitant drugs in the context of ICIs, and previous studies have indicated that patients who received corticoids prior to ICIs treatment exhibit lower OS, but no differences in PFS were observed. [19,20,21]. Our results indicate that when integrating different features in the multivariate analysis, patients who received corticoids did not have statistically significant worse OS, whereas in the univariate analysis they did. This result indicates that use of corticoids may be confounded by other features. Patients who received corticoids exhibited significantly more CNS metastases, and had higher levels of LDH and worse ECOG performance status (Appendix A). Thus, the use of corticoids might reflect the “general status” of a patient, because this treatment in many cases is prescribed to alleviate cancer-related symptoms. In-depth analysis of the different clinical cases of our cohort revealed that this was the case for most patients. None of the other concomitant medications analyzed (beta blockers or anti-diabetic drugs) showed a statistically significant association with OS or PFS in our cohort in any of the models used (multivariate or univariate). 

Our analysis encompassed a wide range of other clinical features. irAEs—regardless of grade—were strongly associated with positive response to ICIs, either considering PFS or OS. This result is consistent with previously published studies [10,11,12,13]. Worse performance status was associated with decreased efficacy of ICIs considering PFS and OS; this result is similar to previous observations [5,7] and consistent with clinical trials [22,23,24]. 

Whether the presence of metastasis in certain organs prior to ICIs treatment is an indicator of limited efficacy of ICIs remains elusive. Our results indicate that the presence of bone metastasis has a negative effect on the response to ICIs (both in terms of OS and PFS), and this observation aligns with recent studies [25]. In contrast, the presence of liver metastasis was only associated with lower OS, which is similar to previous observations [5], but not with decreased PFS. This might indicate that liver metastasis is prognostic rather than predictive of ICIs response. Finally, CNS metastases did not appear to be associated with decreased PFS or OS, in contrast with other series [26]. 

Interestingly, females had statistically significant extended PFS compared to males, and although the impact on OS did not reach statistical significance, it was nearly significant (*p* = 0.07). Nevertheless, the association between sex and PFS was statistically significant only in the multivariate analysis, which may suggest that sex associates with other clinicopathological features included in the model. Thus, further studies may be required to assess the effect of sex in the response to ICIs treatment. Of note, some studies have suggested that females have higher levels of immune infiltration and others have described a larger benefit from the addition of chemotherapy to ICIs in women [27,28].

PD-L1 assessment in our cohort was heterogeneous because some patients had no PD-L1 determination, whereas some had qualitative determination and in others the determination was quantitative. Nonetheless, negativity for PD-L1 was associated with worse clinical outcomes with ICIs.

Notably, packs-year was associated with increased PFS, whereas smoking status was not. As packs-year offers a quantitative assessment of the intensity and duration of the smoking habit, it is possible that this feature better correlates with higher TMB [29,30], which is known to be associated with ICIs response [1,2,3,4]. 

As our cohort is representative of a real-world clinical setting, it encompassed several histology types. Histology does not appear to have a significant prognostic value in our cohort; however, we did observe that Large Cell Neuroendocrine Carcinoma patients had decreased PFS in the univariate analysis, although only a few patients with this histology type were included. Nevertheless, we did not observe differences in OS, similar to the findings of a recent report [31]; hence, the role of ICIs in this histology remains elusive. 

The remainder of the features analyzed did not associate with clinical outcome to ICIs. Notably, elderly patients did not exhibit a worse outcome than other patients. 

Importantly, and in contrast to other studies, we carried out an integrative analysis of multiple features, which allowed us to study their association with clinical benefit, and to investigate their putative interactions. Due to the fact that we found multiple correlations between features, we developed an unbiased methodology to determine the features that might better predict the efficacy of ICIs treatment. As a result, we found that the combination of LDH levels, irAEs, and gender offered the best prognostic value among all of the analyzed features. 

Our study has several limitations. Although our cohort was large, it was based on a single institution. Additionally, our study was retrospective, observational, and descriptive, and aimed at associating different clinical variables with ICIs clinical outcomes; however, it does not provide experimental data of the mechanisms of action behind our observations. Additionally, because we do not have a non-ICIs-treated control, it is difficult to determine whether our findings are predictive or prognostic. Nevertheless, some features are intrinsically associated with ICIs, such as irAEs. Thus, the compendium of features developed is likely to be predictive of ICIs and its prognostic value would be impossible to determine. It is also worth noting that irAEs appear after the treatment has been initiated, and therefore this compendium could not be used for treatment selection. Thus, we also developed a second compendium of features following the same procedure as in Figure 2 but without including irAEs. As a result, the combination of LDH, sex and ECOG performance status appeared to be the best combination of features (cross-validated c-index = 0.635). 

Unfortunately, we did not have accurate expression of PD-L1 for many patients because most of them were treated before its systematic assessment. Hence, we may have underestimated its value and it would be worth exploring in the future whether this could improve our compendium. Similarly, TMB, one of the most promising molecular biomarkers of benefit [1,2,3] and long-term benefit [4] in NSCLC was not available for the vast majority of our cohort and we could not evaluate its importance. It would be of significant interest to add TMB as an additional feature to the models and analyses carried out in this study. This could become a reality as genetic testing (with gene panels or whole exome sequencing) is further incorporated into the clinical setting.

## 5. Conclusions

We integrated multiple clinicopathological features and studied their interplay and value as biomarkers of clinical efficacy of ICIs. From all of the features analyzed, the most prominent associated with positive response (both in terms of PFS and OS) was the appearance of irAEs. Conversely, bone metastasis, and higher levels of LDH and ECOG performance status, were negative indicators of response to ICIs (both in PFS and OS). Our study also indicated strong interactions existed between some features. Thus, we devised a compendium of features (LDH, irAEs, and gender) with additive prognostic value; however, a validation in a prospective cohort is required.

## Figures and Tables

**Figure 1 cancers-13-03249-f001:**
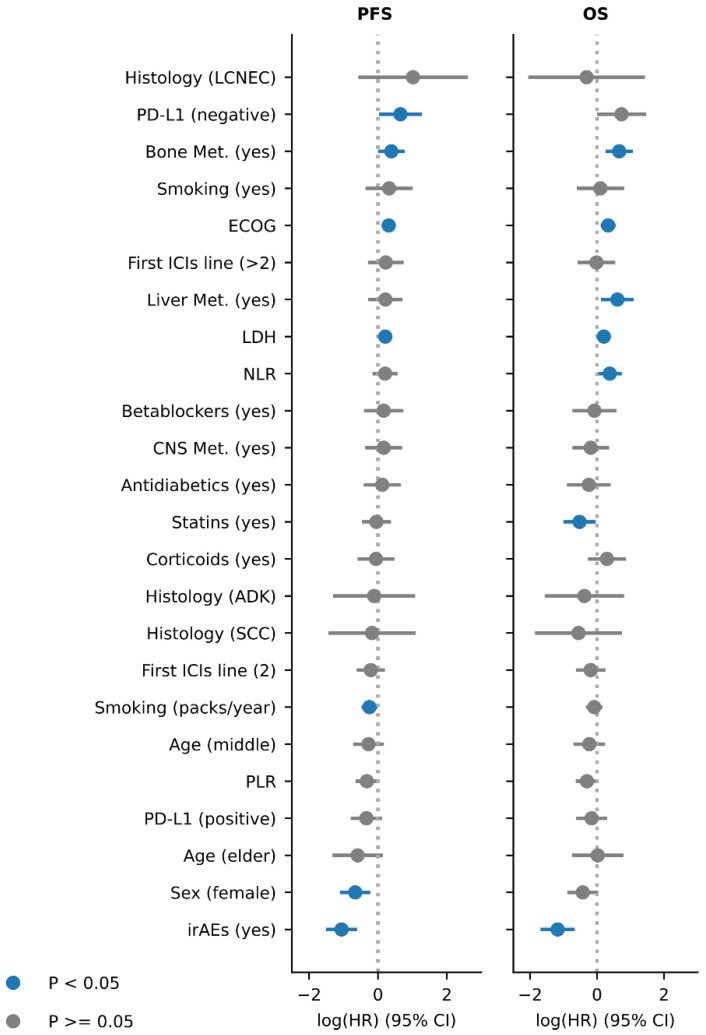
Clinicopathological features associated with ICIs response in multivariate survival models. (**A**) Multivariate Cox proportional-hazards survival model using the PFS as measure of ICIs benefit. (**B**) Multivariate Cox proportional-hazards survival model using the OS as measure of ICIs benefit. Features in blue are statistically significant (*p* < 0.05).

**Figure 2 cancers-13-03249-f002:**
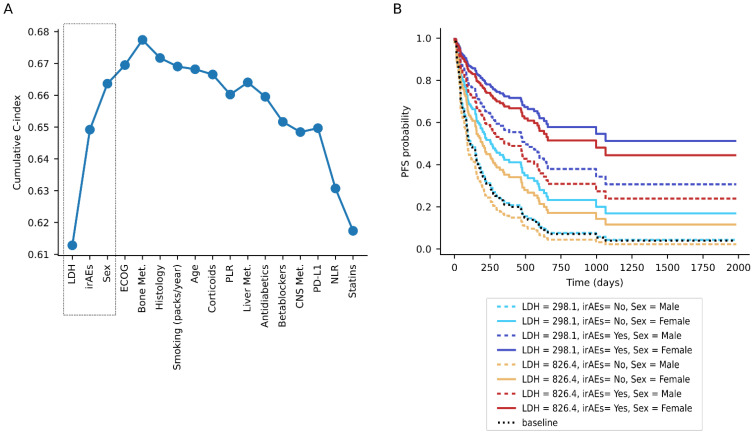
Compendium of features to predict ICIs benefit by maximizing non-redundant information. (**A**) Cumulative concordance index of multivariate Cox proportional-hazards models in which features have been iteratively added. The dotted rectangle encompasses the three features selected to construct the compendium. (**B**) Partial effects representation of the effect of varying the value of LDH, irAEs, or sex in a multivariate Cox proportional-hazards survival model constructed with these three features.

**Table 1 cancers-13-03249-t001:** Patients’ clinicopathological features summary.

**VARIBLE**	**Number of Pts**
**ECOG at ICIs start**	
1	190
0	53
2	18
not available	1
**Histology**	
adk	191
scc	58
nos	7
lcnec	6
**Sex**	
male	182
female	80
**Age category**	
middle	173
young	55
elder	34
**Smoking**	
yes	221
no	30
not available	11
**CNS Met. At ICIs start**	
no	203
yes	58
not available	1
**Liver Met. At ICIs start**	
no	198
yes	63
not available	1
**Bone Met. At ICIs start**	
no	167
yes	94
not available	1
**Corticoids pre-ICIs**	
no	211
yes	45
not available	6
**Statins**	
no	177
yes	85
**Antidiabetic drugs**	
no	222
yes	40
**Betablockers**	
no	227
yes	35
**ICIs line**	
2	101
1	90
>2	53
not available	18
**Treatment type**	
ICIs	203
CT+ICIs	40
TKI+ICIs	19
**PD-L1**	
not available	174
positive	68
negative	21
**Immune-related Adverse Events**	
no	195
yes	67
**VARIABLE**	**Value**
**Smoking (packs/year)**	
Patients with data	251
mean	40.33
std	29.69
min	0.00
25%	20.00
50%	35.00
75%	54.00
max	200.00
**LDH before ICIs start**	
Patients with data	229
mean	589.02
std	692.50
min	211.00
25%	354.00
50%	441.00
75%	591.00
max	7950.00
**Neutrophils to lymphocytes ratio**	
Patients with data	262
mean	5.71
std	6.85
min	0.10
25%	2.41
50%	3.69
75%	6.49
max	72.00
**Platelets to lymphocytes ratio**	
Patients with data	262
mean	249.95
std	204.86
min	15.89
25%	145.18
50%	200.00
75%	292.08
max	2160.00

Pts: patients, ADK: adenocarcinoma, SCC: squamous, NOS: not otherwise specified, LCNEC: large cell neuroendocrine carcinoma, Met: metastases, ICIs: immune checkpoint inhibitors, std: standard deviation, CT: chemotherapy, TKI: tyrosine kinase inhibitor.

**Table 2 cancers-13-03249-t002:** Multivariate Cox proportional-hazards survival model—PFS.

	coef	exp(coef)	coef Lower 95%	coef Upper 95%	exp(coef) Lower 95%	exp(coef) Upper 95%	*p*
irAEs (yes)	−1.058123091	0.347106687	−1.510824435	−0.605421746	0.220727927	0.54584417	4.62473 × 10^−6^
Sex (female)	−0.65991903	0.516893185	−1.099535233	−0.220302828	0.333025827	0.80227581	0.003259403
Age (elder)	−0.590496219	0.554052285	−1.323552824	0.142560386	0.266187901	1.153222717	0.114381452
PD-L1 (positive)	−0.33701656	0.713897018	−0.79189525	0.11786213	0.452985461	1.125088984	0.146467095
Platelets to lymphocytes ratio	−0.322180026	0.724567738	−0.652891336	0.008531284	0.520538547	1.008567779	0.056210093
Age (middle)	−0.276002656	0.758810916	−0.722468713	0.170463402	0.485552087	1.185854251	0.225650961
Smoking (packs/year)	−0.247906764	0.780432704	−0.476616326	−0.019197202	0.6208807	0.98098589	0.033630231
First ICIs line (2)	−0.208104308	0.812122321	−0.617793	0.201584384	0.539132992	1.223339463	0.319455347
Histology (SCC)	−0.174563429	0.839823584	−1.443825478	1.094698621	0.236023128	2.988281943	0.78750135
Histology (ADK)	−0.11182882	0.894197313	−1.305146045	1.081488405	0.271132935	2.949065689	0.854269791
Corticoids pre-ICIs (yes)	−0.058363841	0.943306672	−0.594921579	0.478193898	0.551605821	1.613158241	0.831175452
Statins (yes)	−0.045064475	0.955935846	−0.469399658	0.379270708	0.625377596	1.461218546	0.835112989
Antidiabetic drugs (yes)	0.124236215	1.132283302	−0.417358939	0.665831369	0.658784414	1.946107783	0.653002448
CNS Met. At ICIs start (yes)	0.165526565	1.18001431	−0.369831564	0.700884695	0.690850685	2.015535052	0.544515894
Betablockers (yes)	0.166015388	1.180591269	−0.405754355	0.737785132	0.666473863	2.091298431	0.569300122
Neutrophils to lymphocytes ratio	0.206942426	1.229911758	−0.157958907	0.571843759	0.85388487	1.771530317	0.266339197
LDH before ICIs start	0.212285577	1.236500951	0.033498548	0.391072605	1.034065943	1.478565861	0.019954947
Liver Met. At ICIs start (yes)	0.216398756	1.241597375	−0.282048877	0.71484639	0.754236819	2.043872699	0.394819692
First ICIs line (>2)	0.226301968	1.253954263	−0.292801411	0.745405348	0.746170307	2.107295451	0.392860157
ECOG at ICIs start	0.31212435	1.366324584	0.107503857	0.516744843	1.113495155	1.676561287	0.002792578
Smoking (yes)	0.323985338	1.382627035	−0.36367997	1.011650646	0.695113616	2.750136773	0.355791634
Bone Met. At ICIs start (yes)	0.388614778	1.474936265	0.00245578	0.774773777	1.002458798	2.170101144	0.048560724
PD-L1 (negative)	0.651155188	1.917754917	0.028236571	1.274073805	1.028639002	3.575388368	0.040480882
Histology (LCNEC)	1.016874796	2.764541493	−0.570709179	2.604458771	0.565124522	13.5239038	0.209337005

**Table 3 cancers-13-03249-t003:** Multivariate Cox proportional-hazards survival model—OS.

	coef	exp(coef)	coef Lower 95%	coef Upper 95%	exp(coef) Lower 95%	exp(coef) Upper 95%	*p*
irAEs (yes)	−1.17629018	0.308420805	−1.689034653	−0.663545707	0.184697735	0.515021976	6.9125 × 10^−6^
Sex (female)	−0.425381431	0.653520465	−0.886418215	0.035655353	0.412129271	1.036298628	0.070546507
Age (elder)	0.020976826	1.021198387	−0.748568714	0.790522367	0.473043129	2.204547708	0.957392402
PD-L1 (positive)	−0.162555299	0.849969087	−0.628550091	0.303439493	0.533364571	1.354509632	0.49416206
Platelets to lymphocytes ratio	−0.301424347	0.73976379	−0.635059432	0.032210738	0.529903994	1.032735119	0.076604011
Age (middle)	−0.230925221	0.793798823	−0.704497311	0.242646869	0.49435702	1.274618438	0.339210729
Smoking (packs/year)	−0.083313356	0.920062795	−0.334292255	0.167665543	0.715844543	1.182541035	0.515293654
First ICIs line (2)	−0.190668303	0.826406659	−0.631537143	0.250200537	0.53177376	1.284282938	0.396632245
Histology (SCC)	−0.55449072	0.574364699	−1.853459843	0.744478403	0.15669409	2.105343008	0.402789624
Histology (ADK)	−0.378308731	0.685018982	−1.565201451	0.80858399	0.209045896	2.244727179	0.53215713
Corticoids pre-ICIs (yes)	0.297373388	1.346317904	−0.27292365	0.867670425	0.7611509	2.381356839	0.306782927
Statins (yes)	−0.521778936	0.593463874	−1.00774932	−0.035808552	0.365039643	0.96482499	0.035345032
Antidiabetic drugs (yes)	−0.245060142	0.782657466	−0.900516883	0.410396599	0.406359565	1.507415506	0.4636894
CNS Met. At ICIs start (yes)	−0.188443944	0.82824693	−0.738623387	0.361735499	0.477771169	1.435819117	0.502020309
Betablockers (yes)	−0.077242336	0.925665504	−0.736352462	0.58186779	0.478857384	1.789377494	0.818331261
Neutrophils to lymphocytes ratio	0.379749171	1.461917852	0.020283753	0.739214589	1.020490866	2.094289989	0.038399938
LDH before ICIs start	0.203631208	1.225845988	0.013387751	0.393874665	1.013477768	1.482714701	0.03591487
Liver Met. At ICIs start (yes)	0.609518258	1.839545	0.11918694	1.099849576	1.126580502	3.00371416	0.014835056
First ICIs line (>2)	−0.016989338	0.983154167	−0.579587935	0.545609259	0.560129129	1.725659435	0.95280321
ECOG at ICIs start	0.332175729	1.393997792	0.109309707	0.55504175	1.115507778	1.742013713	0.003486043
Smoking (yes)	0.099953185	1.105119181	−0.610449023	0.810355393	0.543106947	2.24870702	0.782727943
Bone Met. At ICIs start (yes)	0.662856619	1.940327201	0.25418509	1.071528149	1.289410439	2.919838039	0.001477737
PD-L1 (negative)	0.732340436	2.079942889	−0.00562912	1.470309992	0.994386694	4.350583576	0.051773332
Histology (LCNEC)	−0.311295214	0.732497601	−2.056409813	1.433819386	0.127912376	4.194689775	0.72662371

## Data Availability

The cohort level data presented in this study are available in this article, however individual values per patient will be available on request from the corresponding author.

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
