# Peer review of "Interrelations between Patients’ Clinicopathological Characteristics and Their Association with Response to Immunotherapy in a Real-World Cohort of NSCLC Patients"

_cancers, 2021, doi:10.3390/cancers13133249_

Round 1

Reviewer 1 Report

General comment

The present study demonstrated the association between clinicopathological factor such as ECOG PS, LDH, PD-L1, and irAEs and survival after the initiation of ICI therapy in NSCLC. However, above information have been already reported in previous clinical studies. This study may lack novelty. On the other hand, it is valuable to investigate a wide range of patient background factors including medications and to evaluate the relationship between each background factor.

Major comments

  1. The authors raised the issue of the existence of ineffective cases of ICI therapy, and conducted this study for the purpose of predicting the effectiveness of ICI therapy. Several studies have reported the association between irAE and survival and it is reasonable that the association was demonstrated in the present study. However, in order to select patients who would respond to ICI therapy in clinical settings, it is necessary to predict from the status before the initiation of ICI therapy. Is it reasonable to include the occurrence of irAE in the explanatory variables?

  1. The association between treatment difference of ICI monotherapy, ICIs + chemotherapy, and ICIs + tyrosine kinase inhibitors and survival should be analyzed. Combination therapy of ICIs plus chemotherapy might yields longer survival as compared to ICI monotherapy. Furthermore, because it have been reported that survival after the initiation of ICI therapy was short in patients with EGFR mutant NSCLC, the association between the presence of driver mutation and survival after the initiation of ICI therapy should be analyzed.

Minor comments

  1. In Methods, it is described that “stage at diagnosis and metastases localization were monitored (bone, liver and central nervous system were recorded)”. It should be stated that metastatic organ was evaluated at initiation of ICI therapy, not at initial diagnosis.

  1. In Methods, authors should state how PD-L1 expression was evaluated. If PD-L1 expression was evaluated with immunohistochemistry, it should be explained which antibody was used.

  1. In Cox regression hazard model, PD-L1 positive and PD-L1 negative were selected as independent variables. It is presumed that PD-L1 “nan” was selected as reference, and hazard ratio (HR) of both PD-L1 positive and negative versus PD-L1 “nan” was calculated. Instead of such an analysis, PD-L1 negative or PD-L1 positive should be selected as reference, and HR of PD-L1 positive versus PD-L1 negative should be presented.

  1. Table 1 should be prepared in typical style, and explanation of abbreviation should be added. It is unclear what “nan” means.

  1. I’d like authors to discuss about the mechanism of the association between bone metastasis and survival in more detail.

  1. In discussion, it is described that “as packs-year offers a quantitative assessment of the intensity and duration of the smoking habit, it is possible that this feature better correlates with higher TMB”. Please add the explanation whether the evidence of the speculation have been reported.

Author Response

We thank reviewer’s number 1 comments and we address them in blue below.

Reviewer 2 Report

Dear authors, I have read with great interest your study titled "interrelations between patients' clinicopathological characteristics and their association with response to immunotherapy in a real-world cohort of NSCLC patients" and thank you for the opportunity to have it reviewed. The court of enrolled patients is certainly noteworthy, even if about 260 patients in my opinion are not yet a sufficient number to give us such important information that we can draw definitive conclusions especially if with data collected retrospectively. I really appreciated the statistical effort you put into trying to get useful results. Many of the factors that you have shown affect PFS and OS are factors already described in older studies based on more conventional therapies such as gender, LDH, steroids etc. In addition, it is true that you have centered everything on the word "real world" which, however, in my opinion is not very clear, because the other studies on immunocheck points are not real patients? I don't know .... I find this diction unscientific, but obviously this is a pure personal thought. The main problem of analyzing this population of the real world is the sometimes very marked heterogeneity, you also underline it for example that the methods of investigation of the PDL-1 are different, and so many other things, such as the associated therapies. it is clear that these patients will probably have been in more complex and very different therapeutic protocols. In light of all this, I personally do not find that this study does not add information that is so different from what we already knew or that it can help guide therapeutic choices in the near future. I still find it difficult to evaluate the efficacy of immunotherapy outside of prospective randomized studies in well-selected categories of patients, otherwise we risk obtaining somewhat confused information that will not adequately guide us in choosing which therapy for which patient. I believe that if we wanted to obtain more detailed information from this population of patients, it would be necessary to homogenize it more by removing confounding factors and using more "pure" variables that do not include other variables analyzed in the same way, methods such as propensity match or others, it could reduce the number of patients that can be enrolled but could give us more significant results.

Author Response

We thank reviewer’s number 2 comments and we address them in blue below.

Round 2

Reviewer 1 Report

Authors have replied to my comments appropriately.

Reviewer 2 Report

Thank you for the reply